# Exploring Livelihood Strategies of Farmers and Herders and Their Human Well-Being in Qilian Mountain National Park, China

**Jiatong Li [1,2], Haiping Tang [1,2,*] and Foyuan Kuang [3]**

[1] State Key Laboratory of Earth Surface Processes and Resource Ecology (ESPRE), Beijing Normal University, Beijing 100875, China
[2] School of Natural Resources, Faculty of Geographical Science, Beijing Normal University, Beijing 100875, China
[3] School of Economics, Nanjing University of Finance and Economics, Nanjing 210023, China
[*] Correspondence: tanghp@bnu.edu.cn

**Abstract:** National parks have implemented restrictive measures on human activities to protect the ecological environment, which has changed the livelihood dependence and strategies of farmers and herders. Exploring the impact of livelihood strategies on the human well-being of farmers and herdsmen within the national park construction area can help to better develop livelihood intervention measures to achieve sustainable livelihoods for farmers and herders. A total of 329 farmers and herders in Qilian Mountain National Park in China were investigated, and one-way ANOVA and ordinary least squares were employed to analyze the impact of farmers' and herders' livelihood strategies on human well-being in different periods of national park construction. Our results show that the livelihood strategies of farmers and herders play an important role in their well-being. Farmers and herders adopted diversified livelihood strategies after the National Park System Pilot officially launched (after 2015). The well-being advantage of adopting a diversified livelihood strategy became evident over time. Specifically, farmers and herders adopting the settlement diversification livelihood strategy were able to better adapt to the development concept of the national park after the national park pilot was officially established (after 2017). However, nomadic, settled agricultural, and pastoral households were always at a disadvantage in terms of well-being. In addition, distance, nationality, gender, and education level were important factors affecting the well-being of farmers and herders. The results of this study are helpful for improving our understanding of the influence of livelihood strategies on the well-being of farmers and herders and the related challenges they face in the construction of national parks.

**Keywords:** livelihood strategy; human well-being; national park; farmers and herders; China





## 1. Introduction

The increase in population and the subsequent rapid increase in human demand for food, energy, and materials have placed pressure on nature, especially biodiversity degradation and ecosystem services reduction [1–4]. Studies show that more than one-third of the world's land surface and nearly 75% of freshwater resources are currently devoted to crop or livestock production, and land degradation has reduced the productivity of 23% of the global land surface [5]. Facing these adverse effects, countries worldwide are increasingly aware of the urgency of biodiversity protection and are taking various measures [6–8]. One of the essential measures is to effectively protect areas [9]. It is widely recognized that balancing biodiversity and human well-being is a critical need to improve the effectiveness of protecting areas [10,11].

However, balancing the relationship between the two is a relatively difficult task. The establishment of protected areas, such as the implementation of the national park system,

will inevitably bring about changes in policies and the environment, and farmers and herders show significant vulnerability when faced with changes in the environment or policies [12,13], the protection and management of an area lead to changes in the way of life of the indigenous farmers and herders living in such areas, such as the loss of land required for cultivation or grazing and changes in livelihood [14,15], and the well-being of these farmers and herders is affected [16,17]. In China, the government announced its first policy framework for establishing nature reserves with national parks as the main body to support the adequacy of current governance and improve the effective protection of China's protected areas [18]. Therefore, under this emerging trend of natural governance, a holistic understanding of the well-being of farmers and herders in protected areas is necessary for the sustainable development of the ecosystem [19,20].

Intervention targeted at rural livelihoods is considered an effective measure to mitigate the negative impact of environmental changes faced by farmers [21,22]. In general, households and communities pursue diverse livelihood strategies to resist changes in their living environment and improve their well-being [23]. On the other hand, studies have also noted that diversification may make households more vulnerable [24]. Conceptually, different countries or regions have different forms of appropriate livelihood strategies due to natural resources, history and culture, political policies, etc. [25,26]. Similarly, changes in the natural and socioeconomic characteristics of an area over time may also lead households to adjust livelihood strategies with the expectation of improving income and well-being [27]. However, the low level of livelihood awareness of farmers and herders often leads to poor adaptation of their livelihood strategies. If livelihood interventions are to be utilized as an effective measures niche for the reduction in vulnerability of farmers and herders, livelihood strategies should be backed up by proper understanding of their characteristics and their roles in human well-being.

Several studies have documented the relationship between well-being and livelihood strategies, but there is a lack of research on the changes in the relationship between a specific policy or environmental change. Peng et al. [28] finds that livelihood diversification is associated with improvements in SHWB for households with low levels of well-being. Gautam et al. [29] show that livelihood diversification would lead to inequality of well-being. Ibrahim (2023) [30] finds that in remote sectors households are actually transiting to low-return livelihoods. In China, there is a limited understanding of which livelihood strategy farmers and herders should choose. Since the establishment of the national park system in China in 2017, there has been little research on the above aspects in the context of various national park pilot projects, especially relevant research on selecting the best livelihood strategy from the perspective of well-being.

Against this backdrop, this paper comprehensively analyzes well-being, explores the impact of multiple livelihood strategies on well-being, and then analyzes the appropriate livelihood strategies and other influencing factors of well-being in Qilian Mountain National Park. The results from this paper can be used by other researchers as well as policy-makers to promote relevant interventions to support adaptation to changes in biodiversity conservation policies and improve the effectiveness of protecting areas.

To answer the research questions mentioned above, Section 2 mainly introduces the study area, outlines the survey design and research methods, and describes the data processing and analysis methods. Section 3 presents the research results, including information on farmers and herders, descriptive statistics and regression results about the relationship between farmers' and herders' well-being and livelihood strategies. Section 4 analyzes the reasons behind the results. Section 5 summarizes the conclusions and proposes potential policy implications.

## 2. Materials and Methods

### 2.1. Study Area

Qilian Mountain National Park is located on the northwestern border of the Qinghai-Tibet Plateau, at the northern foot of the Qilian Mountains where the three major plateaus

of Qinghai–Tibet, Mongolia–Xinjiang and Loess meet. The national park covers an area of 52,000 km$^2$ and lies between 36°47′–39°48′ N and 94°51′–102°60′ E (Figure 1). The sample collection area is the Qinghai area of the national park, which is populated by a total of 115,600 people in 119 village (pastoral) committees in 20 townships and 4 counties and cities in Delingha City, Tianjun County, Qilian County, and Menyuan County. Traditional livestock and crop farming are the main livelihood activities in the area. The annual average precipitation is approximately 400 mm, with a mean annual temperature of 4 °C. At present, the Qinghai area is a key subject of the reform of the national park system, with emphasis on protection and ecological restoration. However, institutional changes also significantly affect the livelihoods of vulnerable farmers and herders, making the contradiction between ecological protection and farmers' livelihoods acute. Therefore, it is of urgent practical importance to improve the adaptability of farmers and herders to the institutional changes caused by the national park pilot.

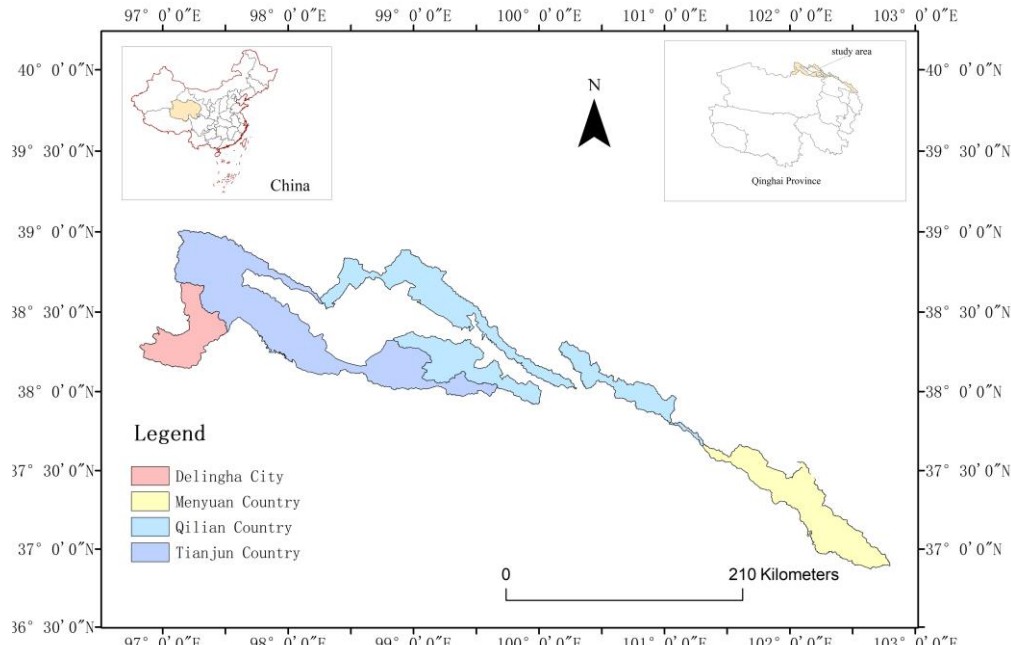

**Figure 1.** Location of the study area.

### 2.2. Survey Design and Data Collection

The questionnaire design was divided into three stages. First, based on papers about farmers' human well-being and livelihood strategy [31], using the Human Well-Being Assessment Framework (Figure 2) from the Millennium Ecosystem Assessment, a preliminary questionnaire was designed. The design and administration of the survey questionnaire followed extensive preliminary qualitative inquiries. Next, five experts in the fields of ecosystem services, human well-being, resource and environmental management, and sociological research, as well as two local government officials, were invited to participate in focused group discussions (FDGs) to obtain opinions on the livelihoods of farmers and herders, as well as human welfare assessment indicators. Based on feedback from the FDGs, the livelihood strategies and human well-being evaluation indicators for farmers and herders wre determined. The livelihood strategies of farmers and herders are divided into six categories: nomadic, settled agricultural, agriculture–pastoral, grazing diversification, settlement diversified, and nonagricultural. The nomadic livelihood strategy takes grazing as its source of livelihood, the agriculture–pastoral type takes agriculture and animal husbandry as its source of livelihood, the settled agricultural type takes agriculture and non-agriculture as its source of livelihood, the grazing diversification type takes husbandry and non-agriculture as its source of livelihood, the settlement diversified type

takes agriculture and non-agriculture as its source of livelihood, and the nonagricultural livelihood strategy takes nonagricultural activities as its source of livelihood.

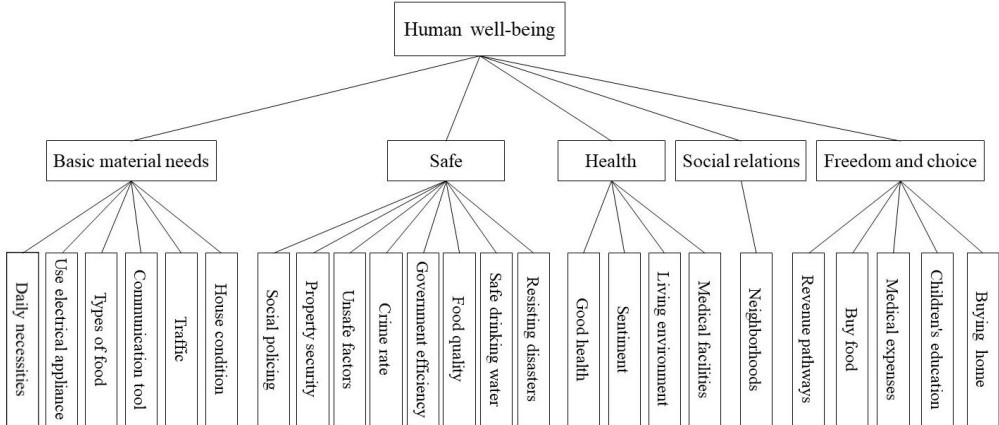

**Figure 2.** The human well-being assessment framework.

A human well-being evaluation index system for farmers and herders from six dimensions (basic material needs, safety, health, social relations, freedom and choice), and corresponding survey questions were also determined. In addition, we set three time nodes, 2015, 2017, and 2021, to show the impact of national park system construction. For the third stage, the preliminary questionnaire was pretested on a sample of 50 local households in May 2021 in Qilian County. Some word order, presentation and logic problems in the questionnaire were identified. The questionnaire was then modified and clarified. The final questionnaire used in the field consisted of three primary parts. The first part includes information on the respondents' age, educational attainment, health status, total number of family members and sources of household income. The second section elicits the feedback of farmers and herders on their own well-being at the three time nodes. The last part focuses on the livelihood strategies farmers adopt to cope with policy changes.

The field survey was conducted in parts of the Qinghai area of Qilian Mountain National Park. Due to the higher sampling accuracy of stratified sampling compared to random sampling, it has been favored by more scholars in practice. Therefore, a sampling technique combining stratified sampling and random sampling is employed to obtain survey data from farmers and herders. We conducted stratified sampling based on the nature of livelihoods, land scale, and population size, with Qilian County, Menyuan County, Delingha City, and Tianjun County selected as the sample counties (cities). Random sampling was used to select the surveyed farmers and herders in the four sample counties (cities), which to some extent made the composition of the surveyed farmers and herders more in line with the overall characteristics of the study area. In rural China, household heads and household agricultural decision-makers are the individuals who have the clearest understanding of household information and play a leading role in household production and management decisions. Therefore, only heads of households or household agricultural decision-makers were allowed to participate in the survey. The members of the survey group were trained to obtain relevant information from the interviewees through face-to-face interviews. Finally, 350 interviews were conducted. After eliminating the responses that lacked information or contained errors, 329 valid survey questionnaires were left for analysis.

*2.3. Data Analysis*

2.3.1. Selection of Well-Being Indicators

Given that the ultimate goal of livelihood diversification for farmers is to improve well-being, assessment dimensions of well-being and indicators for evaluating each dimension remain central to analysis. Because the well-being of farmers and herders involved in national parks is closely related to the ecosystem services provided by national parks [32],

the United Nations Millennium Ecosystem Assessment framework for human well-being derived from ecosystem services was selected. It includes five dimensions to assess overall well-being: basic material needs, safety, health, social relations, and freedom and choice. Each dimension is measured based on indicators identified in the literature specific to the area or similar regions.

Basic material needs are primarily composed of daily necessities, usage of electrical appliances, food abundance, housing quality, transportation and communication, which together enable people to survive in the area and have a vision for future life [33,34]. Safety includes the ability to resist threats to property (money, land) and health (drinking water), which are critical for vulnerable farmers and herders [35]. Health is an important dimension for assessing well-being outcomes, especially physical and mental health. Therefore, physical health and emotions are common indices for evaluating health. Furthermore, a good natural environment and advanced medical facilities can reflect not only the external conditions necessary for health but also people's positive attitude towards their own health. Good social relations refer to the presence of social cohesion and mutual respect. In China, neighborhood relations are an important way to reflect social relations in rural areas. For farmers and herders, declining provision of land has been shown to increase the amount of time needed to collect food and forage grass to satisfy their basic necessities, which in turn reduces the amount of time available for education, employment, and care of family members. Therefore, income pathways, children's education, willingness to make housing purchases, food purchases, and spending on health care were selected as measures of freedom and action.

### 2.3.2. Calculation of Well-Being Score

The indicators for each evaluation dimension have different importance, and each dimension of well-being has different relative importance. Therefore, in the evaluation of well-being, the index weight is an important factor for determining the rationality of the quantitative results. To improve the reliability of the evaluation results, this study attempts to determine the index weight by combining subjective and objective methods [36]. The analytic hierarchy process (AHP) is an appropriate tool for human development projects aiming to improve living standards in developing countries, and it focuses on the needs of beneficiaries [37,38]. The entropy weight method (EWM) based on the entropy information theory can infer useful information from the given data. When the given information of the evaluation index is of great significance, the entropy value is low, and this information should be given a high weight coefficient to indicate its importance. Although the AHP includes the personal preferences and intentions of decision-makers, it has a greater degree of subjectivity and arbitrariness. The EWM will make the evaluation results more objective, but it does not include expert experience or decision-maker opinions. This study used the AHP as the subjective weighting method to determine the specific index weights of five well-being measures. Then, the EWM was used to determine objective weight. Finally, the two values were combined to obtain the final weight. The objective weighting was performed using the following equation [39]:

$$f_{ij} = r_{ij} / \sum_{j=1}^{n} r_{ij}, \, i = 1, 2 \ldots, m; j = 1, 2, \ldots, n$$

where $f_{ij}$ is the ratio of the score $r_{ij}$ of the *j*-th second level indicator of well-being to the i-th first level indicator of well-being, m is the number of primary indicators evaluating well-being, and n is the number of secondary indicators evaluating well-being.

$$H_j = -k \sum_{j=1}^{n} f_{ij} \ln f_{ij} \, , k = 1/ \ln n$$

where $H_j$ is the information entropy of j.

$$W_{ej} = \frac{1 - H_j}{n - \sum_{j=1}^{n} H_j}$$

where $W_{ej}$ is the objective weighting of j. After the values of $W_{ahpj}$ and $W_{ej}$ were calculated, the combined weight of j was obtained using the following equation:

$$W_j = \frac{w_{ahpj} w_{ej}}{\sum_{j=1}^{n} w_{ahpj} w_{ej}}$$

Finally, the well-being score was calculated using the following equation:

$$C_a = \sum_{i=1}^{n} \sum_{j}^{m} W_{ij} r_{ij}$$

where the $C_a$ is the household well-being score of the farmers and herders.

### 2.3.3. Analysis

In the analysis, we use well-being scores as dependent variables and six livelihood strategies as independent variables. In order to improve the credibility of the regression results, based on existing research, this article selects control variables from age, gender, nationality, income, education, and family size [40]. One-way ANOVA is used to explore the differences in household well-being and its indices in the six livelihood strategies that we identified. Ordinary least squares (OLS) has been identified as one of the most popular methods that can be applied to single, multiple, or appropriately coded categorical explanatory variables [41]. OLS is used to study the promotion degree of different livelihood strategies to well-being and significant differences. The data for this study satisfy the linearity assumption of the model [42]. Therefore, the role of six livelihood strategies in household well-being was analyzed using this model. The OLS regression model can be specified as follows:

$$Y_i = \beta_0 + \beta_1 x_1 + \beta_2 x_2 + \ldots + \beta_6 x_6 + \beta_X X + \varepsilon_i$$

where $Y_i$ is the dependent variable, which is the score of the household well-being in pre livelihood strategy; $x_1$ is a vector of independent variables $x_1$, $x_2$, $x_3$, $x_4$, $x_5$ and $x_6$ represent nomadic, sedentary agriculture, agriculture-animal husbandry, sedentary diversification, grazing diversification and nonagricultural livelihood strategies, respectively; $X$ represents the control variable; $\beta_0$ and $\beta_i$ are the vectors of parameters to be estimated; and $\varepsilon_i$ is the error term.

## 3. Results

### 3.1. Socioeconomic Characteristics of the Sample

The main socioeconomic characteristics of the interviewees are reported in Table 1. Overall, the average age of the interviewees was approximately 47 years and ranged from 17 to 79 years. The average educational level of the interviewees ranged from uneducated to junior middle school and was generally low. Ethnically, our survey sample is mainly composed of individuals with Han nationality, Tibetan nationality and Hui nationality. These findings are consistent with the actual situation in Qilian Mountain National Park, where farmers who are engaged in agriculture and animal husbandry generally have the characteristics of middle age, low educational level and complex ethnic composition. The results indicate that the average household size of the interviewed farmers was just over four members. The mean annual net household income of the farmers interviewed was 10,217 USD. In general, the socioeconomic characteristics of the farmers surveyed are basically in line with the actual situation of farmers in Qilian Mountain National Park.

**Table 1.** Socioeconomic characteristics of samples.

| Variable | Variable Description and Assignment | Mean | Std.dev. |
|---|---|---|---|
| Age | Age of the farmers interviewed | 47.03 | 11.28 |
| Education | No formal education = 1; Primary school = 2; Junior middle school = 3; Senior middle school = 4; Junior college and above = 5 | 2.10 | 0.992 |
| Ethnicity | Han = 1; Tibetan = 2; Hui = 3; Others = 4 | 2.06 | 0.905 |
| Hhsize | Household size | 4.22 | 1.515 |
| Livelihood strategies | NS = 1, APS = 2, NAS = 3, GDS = 4, SAS = 5, SDS = 6 | 3.32 | 1.67 |
| Income sources | Number of income sources | 2.31 | 1.75 |
| Distance | Distance from county | 41.14 | 29.10 |

*3.2. Differential Analysis of Human Well-Being under Different Livelihood Strategies*

The ANOVA results show key differences in household well-being and well-being indicator characteristics across the six kinds of livelihood strategies. In household well-being (Figure 3), a significant difference was found between livelihood strategies at the three time nodes. Household well-being is consistently higher among farmers who adopted diversified livelihood strategies. The well-being of farmers and herders who adopted nomadic and settled agricultural livelihood strategies is at a low level. With the construction of the national park, the well-being of farmers and herders who adopted the agriculture–pastoral livelihood strategy ranks third among all livelihood strategies in 2021, showing a high level of well-being.

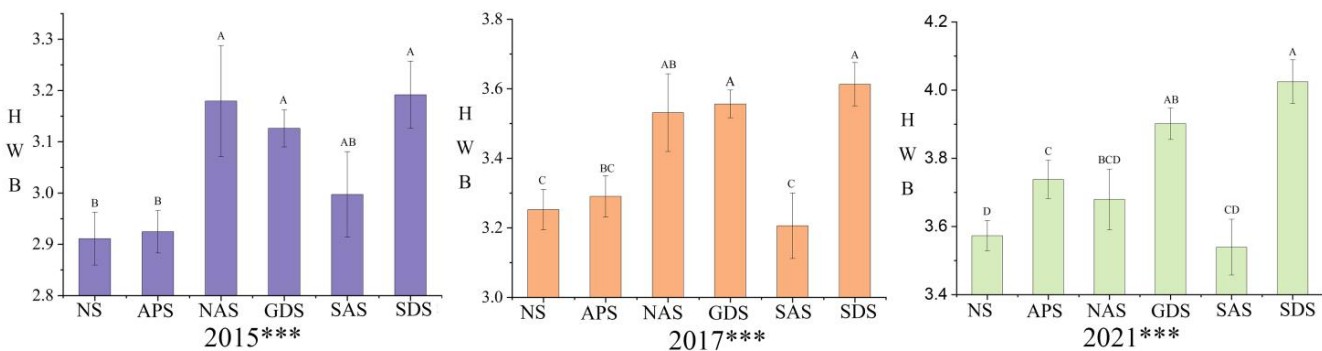

**Figure 3.** The relationship between household well-being and livelihood strategies. Note: NS is the nomadic livelihood strategy; APS is the agriculture–pastoral livelihood strategy; NAS is the nonagricultural livelihood strategy; GDS is the diversified grazing livelihood strategy; SAS is the settled agricultural livelihood strategy; SDS is the settlement diversified livelihood strategy; *** represent significant differences in well-being under different livelihood strategies at the 1% levels. A. AB, C, etc. are obtained from one-way ANOVA, and different superscripts indicate significant differences at the 5% level.

In Figure 4, it is shown that there are significant differences in the basic material needs of different livelihood strategies at various stages of national park construction. The basic material needs of farmers and herders under the diversified livelihood strategy of settlement have always been the highest. After the establishment of the national park pilot program (2021), there was little difference in the basic material level between agricultural and pastoral, diversified grazing, nonagricultural, and settled agricultural livelihood strategies. The basic material level of nomadic livelihood strategies has always been at the lowest level.

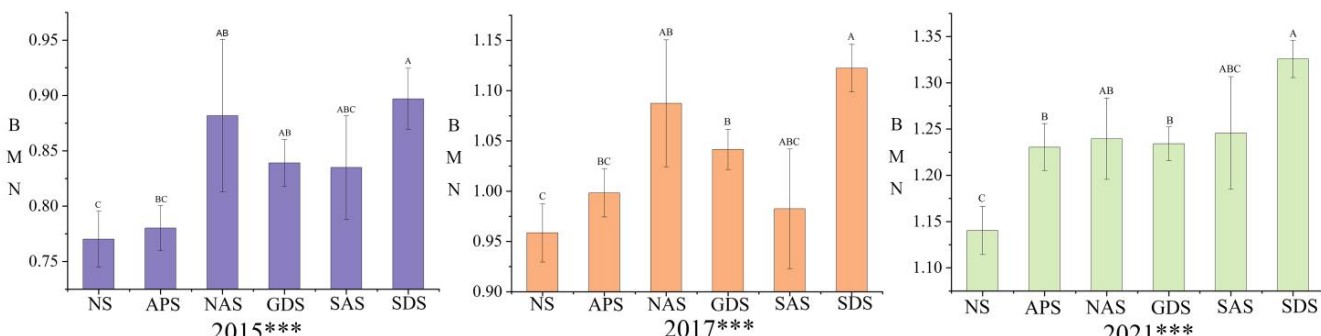

**Figure 4.** ANOVA for basic material needs and livelihood strategies. Note: BMN represents basic material needs. APS is the agriculture-pastoral livelihood strategy; NAS is the nonagricultural livelihood strategy; GDS is the diversified grazing livelihood strategy; SAS is the settled agricultural livelihood strategy; SDS is the settlement diversified livelihood strategy. *** represents significant differences in well-being under different livelihood strategies at the 1% levels. A. AB, C, etc. are obtained from one-way ANOVA, and different superscripts indicate significant differences at the 5% level.

In Figure 5, it is shown that there are significant differences in the safety performance of different livelihood strategies at various stages of national park construction. The safety level of farmers and herders under the diversified livelihood strategy of settlement has always maintained a high level. Before and during the establishment of the national park pilot program (2015 and 2017), the safety level of the nomadic was the lowest, while after the establishment of the national park, the safety level of settled agricultural type was the lowest.

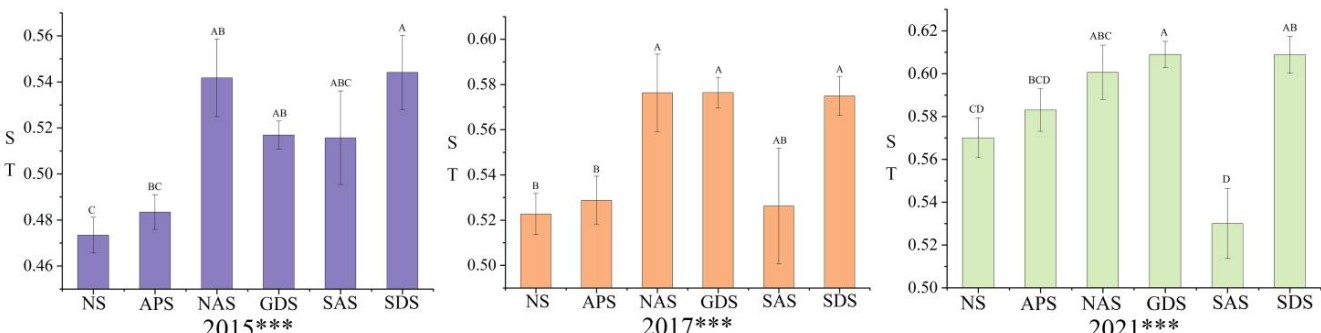

**Figure 5.** ANOVA for safety and livelihood strategies. Note: ST represents safety. APS is the agriculture-pastoral livelihood strategy; NAS is the nonagricultural livelihood strategy; GDS is the diversified grazing livelihood strategy; SAS is the settled agricultural livelihood strategy; SDS is the settlement diversified livelihood strategy. *** represents significant differences in well-being under different livelihood strategies at the 1% levels. A. AB, C, etc. are obtained from one-way ANOVA, and different superscripts indicate significant differences at the 5% level.

In Figure 6, it is shown that there are significant differences in the health performance of different livelihood strategies during and after the establishment of the national park pilot (2017 and 2021). Before the establishment of the national park pilot program (2015), there were no significant differences. The safety of farmers and herders under diversified settlement, diversified grazing, and nonagricultural livelihood strategies has always maintained a high level. The safety level of agriculture and animal husbandry has always been at its lowest.

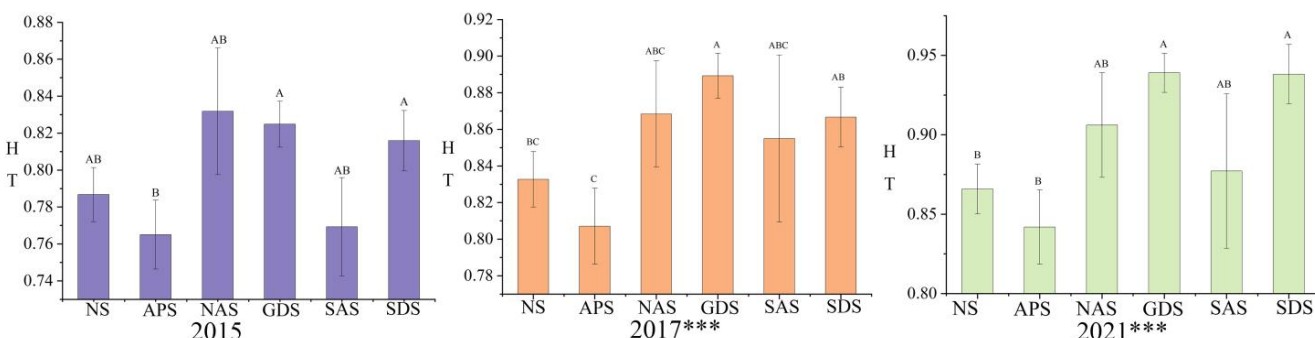

**Figure 6.** ANOVA for livelihood strategies. Note: HT represents health. APS is the agriculture-pastoral livelihood strategy; NAS is the nonagricultural livelihood strategy; GDS is the diversified grazing livelihood strategy; SAS is the settled agricultural livelihood strategy; SDS is the settlement diversified livelihood strategy. *** represents significant differences in well-being under different livelihood strategies at the 1% levels. A. AB, C, etc. are obtained from one-way ANOVA, and different superscripts indicate significant differences at the 5% level.

According to Figure 7, there is no significant difference in the level of social relations under different livelihood strategies during the entire process of pilot construction of national parks. At the time and after the establishment of the national park pilot program (2017 and 2021), the social relationship level of the settlement agricultural livelihood strategy has always been at its lowest, and before the pilot program was established (2015), the social relationship level of the settlement diversified livelihood strategy was at its lowest.

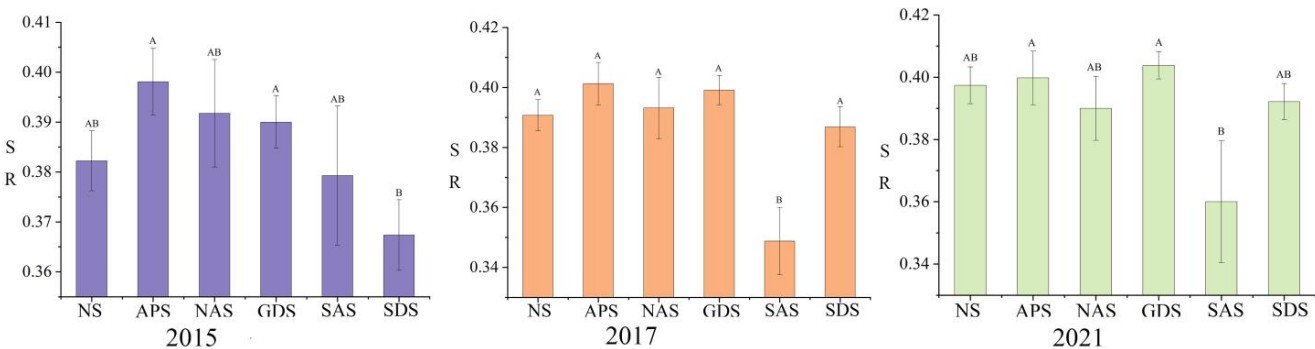

**Figure 7.** ANOVA for social relations and livelihood strategies. Note: SR represents social relations. APS is the agriculture-pastoral livelihood strategy; NAS is the nonagricultural livelihood strategy; GDS is the diversified grazing livelihood strategy; SAS is the settled agricultural livelihood strategy; SDS is the settlement diversified livelihood strategy. A. AB, C, etc. are obtained from one-way ANOVA, and different superscripts indicate significant differences at the 5% level.

In Figure 8, it is shown that there are significant differences in the level of freedom and choice under different livelihood strategies throughout the pilot construction process of national parks. The freedom and choice of settlement and diversified livelihood strategies have always maintained the highest level. Throughout the entire process of pilot construction of national parks, the freedom and choice level of settled agricultural livelihood strategies has always been at its lowest.

At all three time nodes, the mean scores of basic material under the six livelihood strategies are significant, the scores under settlement diversification are always higher than those under the other five strategies, and the scores under the nomadic strategy are lower. After the establishment of the national park, the basic material scores of farmers and herders who settled and diversified also maintain the highest level. When households diversified, their score of security becomes more dominant over time, but the opposite is

the case when households choose a settled agricultural strategy. For health, there is no significant difference between the six livelihood strategies in 2015. The scores of settlement diversification and grazing diversification are higher in 2017 and 2021, but the health score of farmers and herders in the agriculture–pastoral livelihood strategy is always the lowest. The mean score of social relations does not change significantly between the six livelihood strategies. For freedom and choice, diversified strategies are more advantageous than other strategies, and the score of freedom and choice of farmers and herders in the settlement agricultural livelihood strategy is always the lowest with the establishment of the national park.

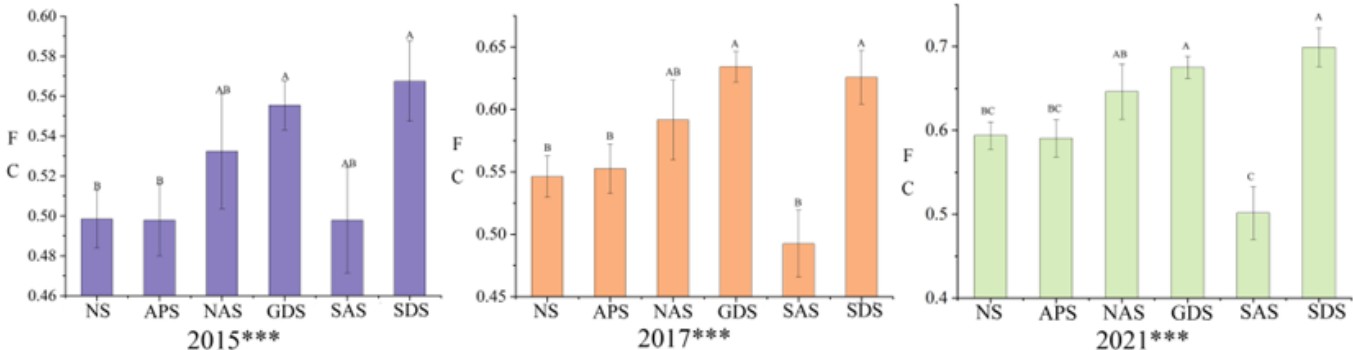

**Figure 8.** ANOVA for freedom and choice and livelihood strategies. Note: FC represents freedom and choice. APS is the agriculture-pastoral livelihood strategy; NAS is the nonagricultural livelihood strategy; GDS is the diversified grazing livelihood strategy; SAS is the settled agricultural livelihood strategy; SDS is the settlement diversified livelihood strategy. *** represents significant differences in well-being under different livelihood strategies at the 1% levels. A. AB, C, etc. are obtained from one-way ANOVA, and different superscripts indicate significant differences at the 5% level.

### 3.3. Farmers' Livelihood Strategies and Human-Well Being

Figure 9 is a scatterplot describing the residuals of the models and explanatory variables (livelihood strategies). This plot demonstrates that our data have homoscedasticity, and this model can be used for statistical inference. Table 2 shows the results for regression estimates predicting the effects of the six different livelihood strategies on farmers' well-being at the three time nodes (2015, 2017, and 2021). Overall, the *p* values of the three time node statistics obtained from the F test are found to be 1%, indicating the models' high goodness of fit and strong explanatory power for dependent variables by the independent variable and control variables inserted in the model. Thus, our results suggest that farmers' choices of livelihood strategies have an impact on their well-being that is significant at the 1% level ($p < 0.001$).

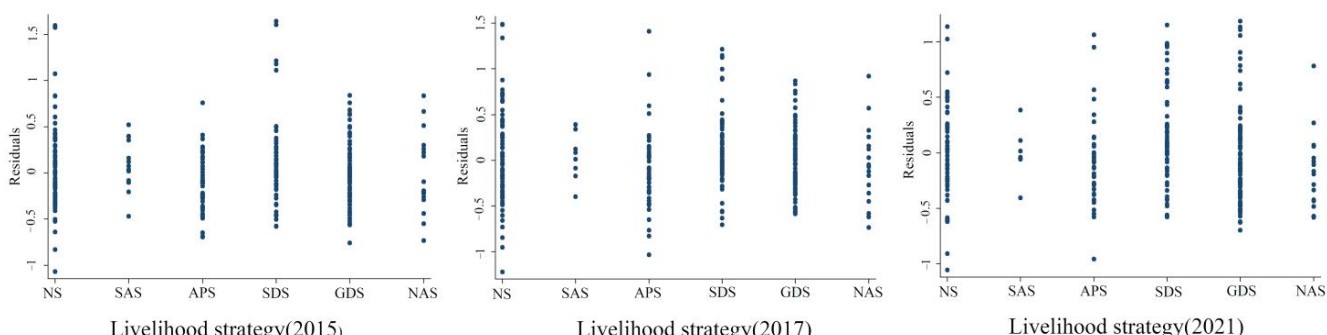

**Figure 9.** Scatterplot of residuals of OLS.

**Table 2.** OLS results for the predicting well-being at the three time nodes.

| Variable | Definition and Assignment | Model 1: Time Node of 2015 | Model 2: Time Node of 2017 | Model 3: Time Node of 2021 |
|---|---|---|---|---|
| SAS | Settled agricultural | 0.034 (0.123) | 0.051 (0.167) | −0.016 (0.173) |
| APS | Agricultural and pastoral | 0.039 (0.080) | −0.005 (0.088) | 0.010 (0.090) |
| SDS | Settlement diversification | 0.231 (0.082) *** | 0.334 (0.083) *** | 0.373 (0.084) *** |
| GDS | Grazing diversification | 0.172 (0.066) *** | 0.266 (0.072) *** | 0.259 (0.070) *** |
| NAS | Nonagricultural | 0.314 (0.130) *** | 0.298 (0.136) ** | 0.262 (0.140) * |
| Distance from county | <15 km = 1; >15 km = 0 | −0.002 (0.001) ** | −0.182 (0.06) * | −0.001 (0.001) |
| Gender | male = 0; female = 1 | −0.056 (0.055) ** | −0.002 (0.001) * | −0.094 (0.056) * |
| Age | ≤45 years = 1; >45 years = 0 | −0.004 (0.048) | −0.071 * (0.059) | 0.030 (0.056) |
| Nationality | Han = 1; Tibetan = 2; Hui = 3; Others = 4 | −0.050 (0.027) * | 0.001 (0.052) * | 0.082 (0.028) *** |
| Income | Number of income sources | −0.056 (0.016) *** | −0.048 * (0.030) | −0.043 ** (0.018) |
| Education | Uneducated = 1; primary school = 2; junior middle school = 3; senior middle school = 4; junior college and above = 5 | 0.078 (0.025) *** | 0.103 (0.028) *** | 0.080 (0.027) *** |
| Hhsize | Household size | −0.024 (0.042) | −0.037 (0.046) | −0.015 (0.044) |
| Income | Total annual net household income | −0.038 (0.025) | 0.057 ** (0.027) | 0.066 ** (0.026) |
| | Constant | 3.102 *** (0.129) | 3.395 *** (0.140) | 3.660 *** (0.133) |
| | Prob > F | 0.0000 *** | 0.0000 *** | 0.0000 *** |
| | Adj R-squared | 0.1120 | 0.1500 | 0.1466 |

Note: Standard errors are in parentheses; * $p < 0.1$, ** $p < 0.05$, *** $p < 0.01$.

Table 2 shows a comparison of the effects on farmers' well-being between the six livelihood strategies. At the three time nodes, compared with the nomadic strategy, the three strategies of settlement diversification, grazing diversification and nonagricultural diversification had the most positive and statistically significant contributions to the well-being of farmers at the 5% level or lower (in 2021, the contribution of the nonagricultural strategy was 10%). Ceteris paribus, the three strategies at the three time nodes increased well-being by more than 0.2 units compared to the nomadic strategy. In particular, for the settlement diversification strategy, the units of relative increase in well-being increased over the years. In 2021, its relative contribution to well-being was the highest at 0.38 units. For the grazing diversification strategy, the relative contribution to well-being units showed a significant increase in 2017 compared with 2015, but after 2017, the change was very weak compared with that of the sedentary diversification strategy.

Diversified livelihood strategies are the main factor that affects the increase in farmers' well-being in Qilian Mountain National Park and surrounding areas. The relative promotion of well-being and the use trend of the settlement diversification strategy indicate that farmers in the study area were more inclined to live a sedentary life. In addition, the difference between the growth trends of the relative contributions of grazing diversification to well-being revealed that the Qilian Mountain National Park pilot has had an impact on farmers who rely on this livelihood strategy.

Table 2 also shows the impact of the eight control variables on farmers' well-being. According to the regression results describing individual and family characteristics, the variable of education reaches a positive impact at the level of 1% ($p < 0.01$), and the variables of gender and ethnicity are significant at the level of 10% or lower. Theoretically, farmers with higher education have more choices of livelihood strategies and can better plan for the diversification of livelihood strategies to improve well-being. Women's participation in livelihood strategies is low in the more backward Qilian Mountain National Park region, and their well-being requirements are lower than those of men, who are the main labor force. In addition, the results showed that in 2015 and 2017, the relative well-being of rural farmers and herders in noncentral counties was relatively low, and this phenomenon was not significant after its establishment. This may be because the construction of national parks has strengthened the construction of remote towns, causing the gap between the well-being of farmers and herders living in them and those living in the county town and surrounding areas to begin to narrow.

## 4. Discussion

For farmers who cultivate food crops or raise cattle and sheep on their lands and rely solely on traditional family farming for subsistence, land, which is key for both farming and grazing, is the most important livelihood resource [43]. However, because the land is located between the atmosphere, biosphere and hydrosphere, it is greatly disturbed by natural and human factors, such as drought [44–46], floods [47–49], and land use [50]. Access to livelihood resources is the key factor leading to vulnerability [51]. Persistent threats to resource access for a long period can cause farmers to face the poverty trap [52], especially smallholders who depend solely on the land for survival. Under the effects of this position, relying on land alone will ultimately depress farmers' well-being [53]. Our findings based on ANOVA are consistent with these studies, indicating that nomadic, settled agricultural and agropastoral households are disadvantaged in terms of well-being.

The results of one-way ANOVA showed that at each time node, the scores of various welfare evaluation indicators corresponding to diversified livelihood strategies were high. Meanwhile, the OLS results indicate that diversified livelihood strategies have the most significant effect on promoting well-being. As such, our results show that diversifying livelihood strategies plays an important role in improving the well-being of farmers, not only overall well-being but also various indicators, such as security, health and freedom and choice. For security, diversification of livelihoods can improve farmers' perception of security [54,55]. This is because the diversification of livelihood strategies is a form of adaptation to fluctuations in resources, seasonality, changes in accessibility and policy [56]. Diversified livelihood strategies can increase nutrient intake, such as food [57], so farmers using these livelihood strategies have better health [58]. Diversified livelihood strategies allow people to have social networks and participate in a set of social interactions to survive and improve their standard of living [59]. This means that farmers adopting these strategies have more channels to learn about knowledge and society, enjoy more opportunities to make choices, and are ultimately more likely to lead a relatively free life.

Our results indicate that the relative average score advantage of most well-being evaluation indicators for farmers engaged in diversified livelihood strategies became more pronounced over time, especially in the pilot phase of Qilian Mountain National Park from 2017 to 2021. This shows that diversified livelihood strategies enable households to better adapt to the concept and sustainable development of national parks in the Qilian Mountains.

Another finding of this study is that, compared with grazing diversification and off-farm employment, the settlement-diverse livelihood strategy most improved the well-being of farmers since the beginning of national park construction in the study area. One possible reason is that in the study area, most of the farmland of the agricultural households that adopted the settlement-diverse livelihood strategy is not located in the core area and is less affected by zoning control. On the other hand, the zoning control of the land in Qilian Mountain National Park makes it harder for households who need more land for grazing in the core area to adapt to policy changes than for agricultural households, who use less land [60]. For example, in interviews, a large number of respondents who used pastoralism as one of their livelihood strategies indicated that they had lost land from the core area for summer pastures. If they felt that these problems could not be addressed by the government, they may become more frustrated and more recalcitrant than agricultural households. This is because herders have a strong sense of belonging influenced by traditional nomadic culture [61].

Finally, although several issues about the impact of livelihood strategies on the well-being of farmers and herders were examined, and we found the livelihood strategies that currently contribute the most to improving well-being to promote sustainable development in Qilian Mountain National Park, this study has some limitations. First, we explored the direct impact of six different types of livelihood strategies on the well-being of farmers and herders but ignored that structural differences across different groups, e.g., age, education, and gender, could have an effect on well-being by affecting livelihood strategies.

Recent studies have shown that basic household characteristics, such as education and livelihood resilience [25], drove well-being by influencing livelihood strategies. Future research can be conducted to test the impact of these different structures to propose specific policy directions for promoting the sustainability of livelihood diversification strategies in national parks. Second, the findings are based on only the Qinghai area of Qilian Mountain National Park and do not involve the Gansu region. Farmers and herders across different regions adopt different adaptation strategies [62] according to the institutional factors, climate, and economic circumstances of the different areas. Thus, much more research must be conducted. Despite the above limitations, the findings of our study contribute to understanding how the adaptation of livelihood strategies may impact farmers' and herders' well-being and support human well-being in the construction of an ecological civilization in China's national parks and nature reserves.

## 5. Conclusions

In this study, we present an integrated well-being analysis framework designed to measure the six forms of livelihood strategies of farmers and herders based on the results of the preinvestigation conducted in the study area. We comprehensively analyze the well-being of farmers and herders, and then the influence of each form of strategy is explored using one-way ANOVA and an OLS regression model. Our study shows that the settlement diversification livelihood strategy, grazing diversification livelihood strategy and nonagricultural livelihood strategy are the primary strategies that improve the well-being of farmers and herders. After the establishment of the Qilian Mountain National Park pilot project, the effect of grazing diversification livelihood strategies in promoting well-being showed a significant upward trend compared with the initial phase, while the settlement diversification livelihood strategies and nonagricultural livelihood strategies did not. Therefore, settlement diversification is the livelihood strategy best adapted to the national park system.

There are several potential policy implications of the results of this study. First, farmers and herders should be encouraged to engage in more off-farm activities, such as trade and management related to national parks. As the well-being of farmers and herders far away from cities is typically at a higher level than that of farmers and herders living near cities, the local government should try to guide rather than force farmers and herders to leave pastoral or agricultural areas to engage in off-farm work in cities. For example, local governments can increase the construction of infrastructure, such as national park structures and roads surrounding them, to increase trade exchanges between households in farming and pastoral areas and cities. Second, we find that the adaptability of the grazing diversity livelihood strategy to the national park system is weaker than that of the settlement diversity livelihood strategy, which has a prominent and positive impact on the well-being indicators of farmers and herders in Qilian Mountain National Park. Consequently, the government should implement measures to solve the difficulties faced by herders, especially the changes after the establishment of the national park pilot. For example, the government can reduce the increase in grazing costs caused by the prohibition of grazing by increasing subsidies for herders to purchase pasture. Third, for farmers and herders, abandoning traditional farming and traditional grazing and then engaging in nonagricultural activities is not the best livelihood option compared to engaging in diversified livelihood activities, especially after the Qilian Mountain National Park pilot was established. Thus, policy interventions for better education of farmers and herders on the impacts of ecological change and adaptation measures in response to ecological change could be enhanced by investing in the provision of training on scientific planting and grazing, which will help promote the sustainable adoption of the national park system.

**Author Contributions:** Conceptualization, J.L. and H.T.; software, F.K.; validation, J.L. and H.T.; data curation, J.L. and F.K.; writing—original draft, J.L.; writing—review and editing, H.T. and F.K.; supervision, H.T.; funding acquisition, H.T. All authors have read and agreed to the published version of the manuscript.

**Funding:** This research was funded by the evaluation and evaluation index system of agriculture and animal husbandry livelihood plan in Qilian Mountain National Park (2021-SF-138) and Research on the Second Comprehensive Scientific Expedition of the Qinghai Tibet Plateau (2019QZKK0606), the funders are Qinghai Provincial Department of Science and Technology and the Second Tibetan Plateau Scientific Expedition and Research Program (STEP).

**Institutional Review Board Statement:** Not applicable.

**Informed Consent Statement:** Not applicable.

**Data Availability Statement:** Not applicable.

**Conflicts of Interest:** The authors declare no conflict of interest.

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
