# Peer review of "Exploring Livelihood Strategies of Farmers and Herders and Their Human Well-Being in Qilian Mountain National Park, China"

_sustainability, doi:10.3390/su15118865_

Round 1

Reviewer 1 Report

Some annotations to the text, mainly (but not wholly) linguistic.

Author Response

Manuscript ID: sustainability-2313060

Title: Exploring livelihood strategies of farmers and herders and their human well-being in Qilian Mountain National Park, China.

Journal: Sustainability

Response to Reviewer 1:

Comment.

  • The peer review comments have been provided in detail in the document "peer review-28126254. v3. pdf ".

Response:

  • Thank you for the detailed comments provided by Reviewer 1, which have been accepted and corrected in the revised manuscript. According to the reviewers’ suggestions and comments, we have conducted language polishing at American Journal Experts ( AJE ), the order number is LKKN3LVB.

Benefited from your critical and constructive reviews, we improved the shortcomings of our paper greatly. Your comments are very significant to improve our manuscript and to guide our future research work. We feel very honored to meet such a good reviewer. Again, thank you very much for reading our article word for word.

Reviewer 2 Report

The work is very interesting, it has merit and relevance.

The same after contributions. 

Additional Comments: 

1. What is the main question addressed by the research?

Exploring livelihood strategies of farmers and herdmen and their human well-being.

2. Do you consider the topic original or relevant in the field? does it
address a specific gap in the field?

Yes, because it brings contributions to the study area, especially from the point of view of sustainability.

3. What does it add to the subject area compared to other published material?

I believe that the work has merit.

4. Are the references appropriate?

Yes.

Author Response

Manuscript ID: sustainability-2313060

Title: Exploring livelihood strategies of farmers and herders and their human well-being in Qilian Mountain National Park, China.

Journal: Sustainability

Response to Reviewer 2:

Comment.

  • Update References.

Response:

  • As your suggestion, we have updated the references in the revised manuscript.

Benefited from your critical and constructive reviews, we improved the shortcomings of our paper greatly. Your comments are very significant to improve our manuscript and to guide our future research work. We feel very honored to meet such a good reviewer. Again, thank you very much for reading our article word for word.

Reviewer 3 Report

Dear Authors,

This manuscript attempts to identify livelihood strategies of human well-beings in Qilian National Park, China. The research has a potential, but the authors should enhance the scientific soundness.

Specific comments are as follows:

1.     Abstract: need to add objective’s statement.

2.     Line 21-25 (also in Introduction section) : when is the establishment of national park, 2015 or 2017? Why introduced piloting national parks in China?

3.     Line 45: need to pay attention on cohesion and coherence in writing the second paragraph, what’s the connection with the first paragraph.

4.     Line 52-53: what are problems associated with farmers? Why should discuss about herdmen, what’s the importance of herdmen? Why there are farming activities in national park?

5.     Line 67-69: from google search, there are several studies have been done in investigating livelihood strategies in Qilian National Park. So, why bother do research the similar topic in the same area again?

6.     Introduction section in general lacks fundamental background of the study literature.

7.     Line 105: Figure 1, the green colour, is it part of the Qilian National Park?

8.     Line 107-108: only one paper?

9.     Line 112: What are the expertise and criteria of these 5 experts?

10.  Line 115-117: 5 or 6 patterns?

11.  Line 123, 133, 136-137: the areas mentioned in these lines should be noted in Figure 1. Country or county?

12.  Line 134: how to do stratified sampling?

13.  Line 177: what are, and the differences between subjective and objective methods?

14.  Line 180: and, what tool used for the objective method?

15.  Line 181: “… denotes them as…” unfinished sentence.

16.  Line 182: any references for widely used in academia?

17.  Line 183: references for equation

18.  Line 184: what are r, m, n?

19.  Line 197: so, what are these six livelihood strategies? Still unclear at all.

20.  Line 199: why use One-way Anova? Why not use the other difference methods?

21.  Line 200-201: what type of OLS regression? Logistic regression?

22.  Line 235: in Figure 3, what the three star (***) is for? What NS, APS, etc stand for? The figure is blurred, not clear.

23.  Line 238: no mentioned anything about settlement in methods section. So how do we know this settlement diversification?

24.  Figures 4 and 5 are blurred as well.

25.  Line 286: where is Table 3?

26.  Line 302: what does it mean in “… between the atmosphere, biosphere and hydrosphere”?

27.  Line 303: do all references cited here about the study area, Qilian National Park?

28.  Line 311: which results support the argument of diversifying livelihood strategies play an important role?

29.  Please improve the overall discussion section with the implications of the research results.

30.  Line 364: what are the integrated analytical framework?

Hope it is useful to improve the quality of the article. Thank you and good luck.

Author Response

Manuscript ID: sustainability-2313060

Title: Exploring livelihood strategies of farmers and herders and their human well-being in Qilian Mountain National Park, China.

Journal: Sustainability

Comment 1.

  • Abstract: need to add objective’s statement.

Response:

  • As your suggestion, we added objective in the abstract ( Line 12-16 ).

Comment 2.

  • Line 21-25 (also in Introduction section) : when is the establishment of national park, 2015 or 2017? Why introduced piloting national parks in China?

Response:

  • Thanks for you point out this problems. The national park pilot was launched in 2015, and the official establishment of the national park pilot in 2017. The establishment of national parks is an effective measure to protect biodiversity.The entire process for the construction of Qilian Mountain National Park have been introduced in the introduction 21-25(.

Comment 3.

  • Line 45: need to pay attention on cohesion and coherence in writing the second paragraph, what’s the connection with the first paragraph.

Response:

  • As your suggestion, we have revised the relevant expressions in the revised manuscript to enhance cohesion and coherence between paragraphs (Line 44-51).

Comment 4.

  • Line 52-53: what are problems associated with farmers? Why should discuss about herdsmen, what’s the importance of herdsmen? Why there are farming activities in national park?

Response:

  • Thanks for you point out this problems. The relevant issue for farmers and herders is to study their livelihood strategies and well-being relationships to improve their vulnerability. Agriculture and animal husbandry are important industries in Qilian Mountain National Park, so we should also study herders. In China, national parks are divided into core areas and general control areas, where livelihood activities are allowed. In lines 46-51, I introduce the effect that national parks can have on the indigenous farmers and herders who live there, by depriving them of arable land or pasture.Comment 5.
  • Line 67-69: from google search, there are several studies have been done in investigating livelihood strategies in Qilian National Park. So, why bother do research the similar topic in the same area again?

Response:

  • We appreciate you point out this problems. Several studies have documents the relationship between well-being and liveli-hood strategies, but there is a lack of research on the changes in the relationship be-tween a specific policy or environmental change. Our paper explores how the process of national park construction affects the relationship between livelihood strategies and well-being. We answer that question in lines 71 to 73.

Comment 6.

  • Introduction section in general lacks fundamental background of the study literature.

Response:

  • As your suggestion, In lines 71 to 77, we added relevant literature background to the research on the relationship between livelihood strategies and well-being .

Comment 7.

  • Line 105: Figure 1, the green colour, is it part of the Qilian National Park?

Response:

  • Taking your opinion into account, we have remade Figure 1 and modified the corresponding expression to make the figure and sentences clearer.

Comment 8.

  • Line 107-108: only one paper?

Response:

  • Sorry, we did not understand this point very well. Line 107-108 is the annotated position in Figure 1.

Comment 9.

  • Line 112: What are the expertise and criteria of these 5 experts?

Response:

  • Taking your opinion into account, we have provided a detailed description of the selection criteria for the five experts (Line 125-129).

Comment 10.

  • Line 115-117: 5 or 6 patterns?

Response:

  • Thanks for you point out this problems. To avoid ambiguity, we have revised the description of the corresponding content in the revised manuscript (Line 129-133).

Comment 11.

  • Line 123, 133, 136-137: the areas mentioned in these lines should be noted in Figure 1. Country or county?

Response:

  • Taking your opinion into account, we have remade Figure 1 and modified the corresponding expression to make the figure and sentences clearer.

Comment 12.

  • Line 134: how to do stratified sampling?

Response:

  • Thanks for your suggestion. We have provided the basis for stratified sampling (Line 149-155).

Comment 13.

  • Line 177: what are, and the differences between subjective and objective methods?

Response:

  • Taking your opinion into account, we have provided a detailed description of the differences between subjective and objective methods in the revised manuscript (Line 199-213).

Comment 14.

  • Line 180: and, what tool used for the objective method?

Response:

  • Thanks for your suggestion. The relevant issues have been revised in the revised manuscript (Line 199-213).

Comment 15.

  • Line 181: “… denotes them as…” unfinished sentence.

Response:

  • Thanks for you point out this problems. Modifications have been made in the revised manuscript (Line 216-217).

Comment 16.

  • Line 182: any references for widely used in academia?

Response:

  • We appreciate you point out this problems, We've removed that expression.
  • Comment 17.
  • Line 183: references for equation.

Response:

  • Taking your opinion into account, We added the reference at line 206.

Comment 18.

  • Line 184: what are r, m, n?

Response:

  • Taking your opinion into account, r is the indictor of well-being, m is the number of primary indicators evaluating well-being, and n is the number of secondary indicators evaluating well-being.

Comment 19.

  • Line 197: so, what are these six livelihood strategies? Still unclear at all.

Response:

  • Taking your opinion into account, six livelihood strategies are nomadic, sedentary agriculture, agriculture-animal husbandry, sedentary diversifica-tion, grazing diversification and non-agricultural livelihood strategies. The livelihood strategies were determined by five experts and two government officials. Before the research, we defined the five livelihood strategies in order to obtain the most accurate livelihood strategies for farmers and herdsmen more accurately.(lines119-133)

Comment 20.

  • Line 199: why use One-way Anova? Why not use the other difference methods?

Response:

  • Taking your opinion into account, One-way ANOVA is used to explore the differences in household well-being and its indices in the six livelihood strategies that we identified.

Comment 21.

  • Line 200-201: what type of OLS regression? Logistic regression?

Response:

  • Taking your opinion into account, The least square method is a continuous variable. logit regression is a discrete variable. Since the dependent variable in this paper is 0-1 continuous data, the least square method is more suitable.
  • Comment 22.
  • Line 235: in Figure 3, what the three star (***) is for? What NS, APS, etc stand for? The figure is blurred, not clear.

Response:

  • Taking your opinion into account, ****, **, * respectively represent significant differences in well-being under different livelihood strategies at the 1%, 5%, and 10% levels.. NS is the nomadic livelihood strategy; APS is the agriculture-pastoral livelihood strategy ; NAS is the non-agricultural livelihood strategy; GDS is the diversified grazing livelihood strategy; SAS is the settled agricultural livelihood strategy; SDS is the settlement diversified livelihood strategy.(lines274-278)

Comment 23.

  • Line 238: no mentioned anything about settlement in methods section. So how do we know this settlement diversification?

Response:

  • Taking your opinion into account, the the settlement diversified type takes agriculture and non-agriculture as its source of livelihood( lines 131-132 ).

Comment 24.

  • Figures 4 and 5 are blurred as well.

Response:

  • Taking your opinion into account, We have sharpened the picture.

Comment 25.

  • Line 286: where is Table 3?

Response:

  • Taking your opinion into account, We changed Table 3 to Table 2.

Comment 26.

  • Line 302: what does it mean in “… between the atmosphere, biosphere and hydrosphere”?

Response:

  • Taking your opinion into account, We describe it this way to show that land is subject to a variety of environmental factors and is unstable.

Comment 27.

  • Line 303: do all references cited here about the study area, Qilian National Park?

Response:

  • Taking your opinion into account, These documents are not all about national parks, they are about land and farmers and herdsmen, because they would lose a lot of land as a result of national park policy, so the purpose of listing these documents is to show that farmers and herdsmen who rely solely on land are vulnerable, to further confirm our results.

Comment 28.

  • Line 311: which results support the argument of diversifying livelihood strategies play an important role?

Response:

  • Taking your opinion into account, Univariate analysis of variance showed that the well-being values of diversified livelihood strategies were all high, and the least square method also showed that diversified livelihood strategies had a significant positive effect on well-being..

Comment 29.

  • Please improve the overall discussion section with the implications of the research results.

Response:

  • Taking your opinion into account, We have made overall changes to the discussion section around the results section.

Comment 30.

  • Line 364: what are the integrated analytical framework?

Response:

Taking your opinion into account, the integrated analytical framework is an integrated well-being analysis framework (Figure 2).

Reviewer 4 Report

The present paper aims at analyzing livelihood strategies of farmers and herdsmen on their well-being in relation to the establishment of the Qilian Mountains National Park pilot.

The overall structure of the paper is good and the argument is well presented, as well as conclusions.

Nevertheless an extensive language revision is necessary,starting from the title (herdmen instead of herdsmen); the same applies to formatting (e.g. keywords should be all in capital letters or viceversa; line 303 has a double spacing and so on).

Author Response

Manuscript ID: sustainability-2313060

Title: Exploring livelihood strategies of farmers and herders and their human well-being in Qilian Mountain National Park, China.

Journal: Sustainability

Comment 1.

  • Nevertheless an extensive language revision is necessary,starting from the title (herdmen instead of herdsmen); the same applies to formatting (e.g. keywords should be all in capital letters or viceversa; line 303 has a double spacing and so on)..

Response:

  • According to the reviewers’ suggestions and comments, we have conducted language polishing at American Journal Experts ( AJE ), the order number is LKKN3LVB. In addition, some formatting issues have also been improved in the revised manuscript.
  • Benefited from your critical and constructive reviews, we improved the shortcomings of our paper greatly. Your comments are very significant to improve our manuscript and to guide our future research work. We feel very honored to meet such a good reviewer. Again, thank you very much for reading our article word for word.

Reviewer 5 Report

The authors have to consider this comments. 

1- Specific suggestions have not been provided.

2- Why is the stratified random sampling technique used for sampling?

3- Research literature should be strengthened. Previous studies have been properly reviewed and the reason for conducting a new study should be mentioned.

4- Page 5, line 198, what is the difference between control variable and independent variables?

5- Why is the OLS method used in estimation?

6- In line 206, instead of showing a general form of the relationship, the experimental model of this research should be specified.

7- In Table 1, all model variables should be introduced and their specifications should be given.

8- What has diagram 5 contributed to the analysis and why is it shown in this section?

9- Why are ANOVA and regression methods used at the same time? How do these two help each other?

10- Why r2 is very low in these models? Can the estimation results of these models be trusted? There is a problem with the model specification.

11- There is uncertainty about the estimation results, and based on this, the discussion and conclusion are not very reliable. The estimated model should be improved, so that r2 increases significantly.

Author Response

Manuscript ID: sustainability-2313060

Title: Exploring livelihood strategies of farmers and herders and their human well-being in Qilian Mountain National Park, China.

Journal: Sustainability

Comment 1.

  • Specific suggestions have not been provided.

Response:

  • Thanks for your suggestion. Specific suggestions have been provided (Line 465-487).

Comment 2.

  • Why is the stratified random sampling technique used for sampling?

Response:

  • As your suggestion, we have revised the annotation positions of the figures and tables in the revised manuscript to make them as suitable as possible.

Comment 3.

  • Line 213: Table should be displayed in capital letters.

Response:

  • We appreciate you point out this problems. We have carefully reviewed the manuscript and made modifications to similar issues as required (Line 261).

Comment 4.

  • Line 236: Why is Figure 3 given? The place of reference in the manuscript is not specified.

Response:

  • Thanks for you point out this problems. Figure 3 depicts the differential analysis of different livelihood strategies and human well-being of farmers and herders during different periods of national park construction. This is a numerical exploration of whether there are differences, which is the basis for subsequent regression analysis. Furthermore, we have revised the annotation positions of the figures and tables in the revised manuscript to make them as suitable as possible.

Comment 5.

  • Line 253: Why is Figure 4 given? The place of reference in the manuscript is not specified.

Response:

  • We appreciate you point out this problems. Figure 4 depicts the differential analysis of different livelihood strategies and human well-being (five evaluation dimensions) of farmers and herders during different periods of national park construction. This is a numerical exploration of whether there are differences, which is the basis for subsequent regression analysis. Moreover, we have revised the annotation positions of the figures and tables in the revised manuscript to make them as suitable as possible.

Comment 6.

  • Line 257, 286: Where in the manuscript is Table 3 often referenced in the manuscript? There is an error in this or Table 3 is not presented in the manuscript.

Response:

  • As your suggestion, this is an error and has been corrected.

Comment 7.

  • Line 236, 253 and 266: The data in the figures (Figure 3, Figure 4 and Figure 5) presented in the manuscript are not read very faintly. It should be presented more clearly

Response:

  • Taking your opinion into account, we will split Figure 4 in the original manuscript into Figures 4-8 to make it clearer.
  • Benefited from your critical and constructive reviews, we improved the shortcomings of our paper greatly. Your comments are very significant to improve our manuscript and to guide our future research work. We feel very honored to meet such a good reviewer. Again, thank you very much for reading our article word for word.

Reviewer 6 Report

The article is well structured. The applied methods are appropriate and have created opportunities to compare the researched strategies of farm households. The methodological part contains the necessary elements - the characteristics of the studied area, the research methods, the tested strategies, the framework for assessing human well-being. The results are presented and discussed from the standpoint of human welfare under different livelihood strategies.

For the editing of the article, I have the following suggestions:

- I recommend presenting the six farm household strategies in more detail. It is not clear from their names which of the strategies include ecosystem services or other specific activities in the national park.

- Readers will also be interested in the specific weight of the index of five well-beings obtained using the AHP weighting method.

- Figure 4 contains 15 graphs of the relationship between welfare indicators and livelihood strategies, which are insufficiently analyzed by the authors of the article.

- Figures 3 and 4 are not referred to in the text.

Author Response

Manuscript ID: sustainability-2313060

Title: Exploring livelihood strategies of farmers and herders and their human well-being in Qilian Mountain National Park, China.

Journal: Sustainability

Comment 1.

  • I recommend presenting the six farm household strategies in more detail. It is not clear from their names which of the strategies include ecosystem services or other specific activities in the national park.

Response:

  • Thanks for your suggestion. In the revised manuscript, we provide necessary descriptions for the determination and definition of six livelihood strategies for farmers and herdsmen. (Line 125-133).

Comment 2.

  • Readers will also be interested in the specific weight of the index of five well-beings obtained using the AHP weighting method.

Response:

  • We appreciate you point out this problems. We have provided a detailed description of the selection basis and determination method for the weight determination of human well-being evaluation indicators (Line 195-226).

Comment 3.

  • Figure 4 contains 15 graphs of the relationship between welfare indicators and livelihood strategies, which are insufficiently analyzed by the authors of the article.

Response:

  • Thanks for you point out this problems. We will split Figure 4 in the original manuscript into Figures 4-8 and provide a more detailed description of each figure.

Comment 4.

  • Figures 3 and 4 are not referred to in the text.

Response:

  • We appreciate you point out this problems. We have revised the annotation positions of the figures 3 and 4 in the revised manuscript to make them as suitable as possible.
  • Benefited from your critical and constructive reviews, we improved the shortcomings of our paper greatly. Your comments are very significant to improve our manuscript and to guide our future research work. We feel very honored to meet such a good reviewer. Again, thank you very much for reading our article word for word.

Reviewer 7 Report

Dear Authors

Agricultural activities, which have a vital role in the protection, maintenance and development of human existence, have a feature that is directly affected by nature and also directly affects nature. With this aspect, the protection of these resources in the process of meeting human needs is of vital importance in terms of sustainable economic activities. The study is important in terms of examining the welfare of both components in the integration of nature and human and producing solutions to possible problems.

Additional Comments:

As a reviewer, I would like to draw your attention to the points I have stated in the table below in your manuscript, believing that each different eye will make the study more valuable.

Point 1

Line 105

Why is Figure 1 given? The place of reference in the manuscript is not specified.

Point 2

Line 109

Tables and Figures should be positioned as close to the sentence they are referred to as a reference.

Point 3

Line 213

Table should be displayed in capital letters

Point 4

Line 236

Why is Figure 3 given? The place of reference in the manuscript is not specified.

Point 5

Line 253

Why is Figure 4 given? The place of reference in the manuscript is not specified.

Point 6

Line 257, 286

Where in the manuscript is Table 3 often referenced in the manuscript? There is an error in this or Table 3 is not presented in the manuscript.

Point 6

Line 236, 253 and 266

The data in the figures (Figure 3, Figure 4 and Figure 5) presented in the manuscript are not read very faintly. It should be presented more clearly

Author Response

Manuscript ID: sustainability-2313060

Title: Exploring livelihood strategies of farmers and herders and their human well-being in Qilian Mountain National Park, China.

Journal: Sustainability

Comment 1.

  • Line 105: Why is Figure 1 given? The place of reference in the manuscript is not specified.

Response:

  • Thanks for your suggestion. The presentation of Figure 1 can better inform readers of the longitude and latitude information of the research area and sampling locations. Figure 1 has been optimized as required and annotated in the manuscript (Line 105).

Comment 2.

  • Line 109: Tables and Figures should be positioned as close to the sentence they are referred to as a reference.

Response:

  • As your suggestion, we have revised the annotation positions of the figures and tables in the revised manuscript to make them as suitable as possible.

Comment 3.

  • Line 213: Table should be displayed in capital letters.

Response:

  • We appreciate you point out this problems. We have carefully reviewed the manuscript and made modifications to similar issues as required (Line 261).

Comment 4.

  • Line 236: Why is Figure 3 given? The place of reference in the manuscript is not specified.

Response:

  • Thanks for you point out this problems. Figure 3 depicts the differential analysis of different livelihood strategies and human well-being of farmers and herders during different periods of national park construction. This is a numerical exploration of whether there are differences, which is the basis for subsequent regression analysis. Furthermore, we have revised the annotation positions of the figures and tables in the revised manuscript to make them as suitable as possible.

Comment 5.

  • Line 253: Why is Figure 4 given? The place of reference in the manuscript is not specified.

Response:

  • We appreciate you point out this problems. Figure 4 depicts the differential analysis of different livelihood strategies and human well-being (five evaluation dimensions) of farmers and herders during different periods of national park construction. This is a numerical exploration of whether there are differences, which is the basis for subsequent regression analysis. Moreover, we have revised the annotation positions of the figures and tables in the revised manuscript to make them as suitable as possible.

Comment 6.

  • Line 257, 286: Where in the manuscript is Table 3 often referenced in the manuscript? There is an error in this or Table 3 is not presented in the manuscript.

Response:

  • As your suggestion, this is an error and has been corrected.

Comment 7.

  • Line 236, 253 and 266: The data in the figures (Figure 3, Figure 4 and Figure 5) presented in the manuscript are not read very faintly. It should be presented more clearly

Response:

  • Taking your opinion into account, we will split Figure 4 in the original manuscript into Figures 4-8 to make it clearer.

Benefited from your critical and constructive reviews, we improved the shortcomings of our paper greatly. Your comments are very significant to improve our manuscript and to guide our future research work. We feel very honored to meet such a good reviewer. Again, thank you very much for reading our article word for word.

Round 2

Reviewer 3 Report

All suggestions have been considered and revised.

Reviewer 5 Report

Drae Editor. 

I'm satisfy about the revised version. 

Regards